# Brain MRI findings in relation to clinical characteristics and outcome of tuberculous meningitis

Sofiati Dian [1,2]*, Robby Hermawan[3], Arjan van Laarhoven[4], Sofia Immaculata[2], Tri Hanggono Achmad[2], Rovina Ruslami[2], Farhan Anwary[5], Ristaniah D. Soetikno[5], Ahmad Rizal Ganiem[1,2], Reinout van Crevel[4,6]

1 Department of Neurology, Faculty of Medicine, Hasan Sadikin Hospital, Universitas Padjadjaran, Bandung, Indonesia, 2 Infectious Disease Research Center, Faculty of Medicine, Hasan Sadikin Hospital, Universitas Padjadjaran, Bandung, Indonesia, 3 Department of Radiology, St. Borromeus Hospital, Bandung, Indonesia, 4 Department of Internal Medicine, Radboud University Medical Center, Nijmegen, The Netherlands, 5 Department of Radiology, Faculty of Medicine, Hasan Sadikin Hospital, Universitas Padjadjaran, Bandung, Indonesia, 6 Centre for Tropical Medicine and Global Health, Nuffield Department of Medicine, University of Oxford, Oxford, United Kingdom

* sofiatidian@gmail.com, sofiati.dian2016@unpad.ac.id

**Data Availability Statement:** All relevant data are within the manuscript and its Supporting Information files (excel file).

## Abstract

Neuroradiological abnormalities in tuberculous meningitis (TBM) are common, but the exact relationship with clinical and inflammatory markers has not been well established. We performed magnetic resonance imaging (MRI) at baseline and after two months treatment to characterise neuroradiological patterns in a prospective cohort of adult TBM patients in Indonesia. We included 48 TBM patients (median age 30, 52% female, 8% HIV-infected), most of whom had grade II (90%), bacteriologically confirmed (71%) disease, without antituberculotic resistance. Most patients had more than one brain lesion (83%); baseline MRIs showed meningeal enhancement (89%), tuberculomas (77%), brain infarction (60%) and hydrocephalus (56%). We also performed an exploratory analysis associating MRI findings to clinical parameters, response to treatment, paradoxical reactions and survival. The presence of multiple brain lesion was associated with a lower Glasgow Coma Scale and more pronounced motor, lung, and CSF abnormalities (p-value <0.05). After two months, 33/37 patients (89%) showed worsening of MRI findings, mostly consisting of new or enlarged tuberculomas. Baseline and follow-up MRI findings and paradoxical responses showed no association with six-month mortality. Severe TBM is characterized by extensive MRI abnormalities at baseline, and frequent radiological worsening during treatment.

## Introduction

Meningitis is the most severe manifestation of tuberculosis (TB), causing death or disability in up to 50% of those affected [1]. *Mycobacterium tuberculosis* is hypothesized to spread during an early bacteraemia phase, leading to granuloma development in all organs including the brain [2]. TB meningitis (TBM) occurs as a result of later rupture of meningeal or para-

**Funding:** This work was supported by Peer Health (National Academy of Sciences [NAS]- United States Agency for International Development [USAID]), the Ministry of Research, Technology and Higher Education, Indonesia (PKSLN grant to T.H.A., R.R., and S.D.), the Direktorat Jenderal Pendidikan Tinggi (BPPLN fellowship to S.D.), and Radboud University Medical Center, the Netherlands (fellowship to S.D.).

**Competing interests:** The authors have declared that no competing interest exist

meningeal granulomatous lesions in the subarachnoid space. Entrapment of penetrating arteries, cranial nerves and the ventricular system, can lead to stroke, cranial nerve palsy and hydrocephalus, respectively [2, 3]. As such, localisation, extent and nature of the inflammatory process probably have a major effect on the clinical manifestations and outcome of TBM.

Imaging plays a major role in detection of brain abnormalities related to TBM. Magnetic Resonance Imaging (MRI) offers a non-invasive soft tissue contrast imaging modality that offers high spatial resolution without ionizing radiation [4]. MRI can facilitate the exploration of the morphological features of the human brain, hence advancing our insights into the neurobiology processes of a disease [5]. MRI has a much higher discriminatory value compared to computed tomography (CT), in particular for detecting most pathologic meningeal conditions [6]. Previous studies using MRI have found the common triad of neuroradiological findings in TBM, i.e. basal meningeal enhancement, hydrocephalus, and infarction [3].

Several factors reassure the image quality of MRI, including signal-to-noise ratio (SNR) related to slice thickness and the patient's ability to remain motionless during examination [7, 8]. The sequence used and reading methods is also important [7]. Gadolinium-enhanced axial T1 two-dimension spin echo (2D-SE), one of the most important sequences, is often used for detecting pathology in the central nervous system (CNS). However, slices thinner than 3 mm are not feasible due to the inferior signal-to-noise ratio, and the subsequent slice gap may result in missing small lesions [9]. Magnetization-prepared rapid gradient-echo (3D MP-RAGE) imaging is a three-dimensional MRI sequence which produces slices with thickness of 1 mm or less, yielding an image with superior tissue contrast and higher spatial resolution compared to 2D-SE [9, 10]. Administration of gadolinium-diethylenetriaminepentaacetic acid (DTPA) allows for visualization of inflamed meninges [11]. We used MRI 3D MP-RAGE to examine the neuroradiological patterns, and explored the possible association of neuroradiological findings to clinical characteristics and outcome in a prospective cohort of clinically well-characterised TBM patients in Indonesia.

## Material and methods

### Setting, patients and follow-up

We included consecutive patients enrolled in a double-blind randomized clinical trial on high dose rifampicin for tuberculous meningitis between December 2014 and June 2016 in Hasan Sadikin Hospital [12], Bandung, Indonesia—the referral hospital for the province of West Java. Patients above 14 years of age with signs and symptoms of tuberculous meningitis, combined with a CSF/blood glucose ratio <0.5 and CSF leukocyte count ≥5, and with negative CSF India Ink and Gram staining were included. CSF microscopy (Ziehl-Neelsen), liquid *M. tuberculosis* culture (MODS), and GeneXpert MTB/RIF were done on large (>5 ml) CSF samples in all patients [13]. A detailed medical history and clinical examination was performed, and blood and CSF inflammatory markers as well as chest X-ray abnormalities were assessed before start of anti-tuberculous drugs.

TBM was treated with a combination of rifampicin, isoniazid, ethambutol and pyrazinamide for at least 6 months. All patients were given adjunctive dexamethasone in a tapering regimen [14]. Two thirds of our patients had one month of double or triple rifampicin dosing as a study drug used in this trial. Patients were followed for 6 months. The clinical outcome was death or severe disability 6 months after inclusion. Clinical improvement after anti-tuberculosis administration was assessed using Glasgow Outcome Scale (GOS). Good recovery defined as score GOS 4 or 5 [15]. In case the patient experienced a symptomatic paradoxical response, we increased the dose of dexamethasone back to the starting dose accordingly to their TBM grade.

Written informed consent to participate in the trial was obtained from all patients or from their relatives if the patient could not provide informed consent. Consent was registered in a REDCap clinical research form for all patients, and the same procedure was followed for patients over 14 and under 18 years of age, who are considered adult according to local custom. All study procedures were performed in accordance with relevant guidelines and regulations. This study and the written informed consent were approved by the ethical review of the Medical Faculty of Universitas Padjadjaran, Bandung, Indonesia, No.330/UN6.C1.3.2/KEPK/PN/2016. This trial is registered with ClinicalTrials.gov as trial number NCT02169882.

## MRI protocol

Baseline brain MRI was performed within 5 days after diagnosis and start of treatment to capture baseline abnormalities, and again after 2 months (plus or minus 1 week) of anti-tuberculous treatment as a study related protocol using the 1.5 Tesla systems (Magnetom Essensa, Siemens Healthcare, USA. The MRI sequences included T1 spin echo, axial T2 turbo spin echo, axial T2 fluid attenuation inversion recovery (FLAIR) and axial diffusion weighted imaging (DWI)—apparent diffusion coefficient (ADC), and axial T2 gradient echo (GRE) with and without contrast media, and axial T1 3D magnetization-prepared rapid acquisition with gradient-echo (MP-RAGE) (isotropic) with and without intravenous contrast media. Post-contrast study was carried out using Gadolinium 0.2 mmol/kg/body weight intravenously. The 3D MP-RAGE is a high-resolution three-dimensional (3D) MRI sequence of 1 mm thick slices resulting in more precise anatomic localization of lesion morphology [10]. All imaging was performed in the axial plane and with identical geometrical parameters. All images were evaluated using OsiriX MD 8.0 for Mac, Pixmeo, SARL, Switzerland.

Systematic assessment of the imaging studies was undertaken and quantified by consensus. Radiologist (RH, FA, RDS) and a neurologist (SD) evaluated all MRIs for the presence of enhancement of the leptomeninges, and presence of hydrocephalus, tuberculomas, brain infarction and cranial nerve enhancement and did not aware to treatment arm and outcome. The location of the infarction was also specified as: cerebrum, cerebellum or brain stem. Cerebral infarction was further categorized into inside or outside the basal ganglia and thalamus. Infarction was categorized as in the basal ganglia if located in the striatum or caudate nucleus, putamen, and globus pallidum [16].

## Paradoxical response

MRIs after 2 months treatment were compared with baseline MRIs. Radiological paradoxical responses were defined as worsening of pre-existing tuberculous lesion or appearance of new lesions in the second brain MRI (**Box 1**), whereas a clinical paradoxical response was defined

---

Box 1. Definition of paradoxical response

General condition

Definite or probable tuberculous meningitis patients with:

1. Initial improvement during anti-TB therapy

2. > 28 days of treatment

3. Compliance to anti-tuberculosis treatment

4. No anti-tuberculosis drugs resistance

Who present with new or worsening clinical or radiological signs.

## Radiological paradoxical response

Development of new or worsening lesions in radiological finding, including:

## Hydrocephalus

New event of hydrocephalus either communicating or non-communication or worsening of pre-existing one; or increasing area of trans ependymal leakage (periventricular edema) from the hydrocephalus.

## Leptomeningeal enhancement

Formation of new leptomeningeal enhancement in other areas of subarachnoid spaces; or increasing (thickening) of the previous leptomeningeal enhancement.

## Tuberculoma

Increasing size of previous tuberculoma; or new or additional formation of tuberculoma at different areas of the brain or subarachnoid spaces; or formation or increasing oedema surrounding the tuberculoma; or increasing mass effect from the tuberculoma such as increasing effacement of sulci and cisterns, increasing midline shift; or new or worsening sign of herniation.

## Infarction

New or additional formation of infarction whether it is new area of acute infarction or new area of chronic infarction. Acute infarction is defined as infarct lesion that gives restricted diffusion appearance on DWI-ADC sequence of the MRI. Chronic infarction is defined as infarct lesion that does not give restricted diffusion appearance on DWI-ADC sequence of the MRI. including the formation of new infarction in other areas of the brain or another vasculatory system with or without the sign of vasculitis

as a new neurological event, including cranial nerve palsy, motor deficits, seizures, or severe headache, with or without the worsening of brain imaging, in patients whose condition initially improved with anti-tuberculous treatment [17]. Clinical-radiological paradoxical responses were defined as the presence of both radiological and clinical paradoxical responses. Paradoxical response was considered as 'definite' if it occurred more than 4 weeks after commencement of anti-tuberculous treatment [17].

### Data analysis and statistics

Patient characteristics were presented as medians or proportions as indicated. Clinical characteristics and six-month survival and functional outcome were compared between patients with or without MRI findings at baseline, and with or without a paradoxical worsening of MRI findings after two-months, using Kruskal-Wallis test for continuous variables and Chi-square test

for categorical variables. All statistical analysis was performed using IBM SPSS Statistics version 24, p-value of <0.05 were considered significant.

## Results

### Baseline MRI findings

Sixty patients (52% female, median age 30.5 years, 8% HIV co-infected) were enrolled in a phase IIb dose-finding clinical trial, 48 had baseline MRIs and were eligible for this study. They had an MRI after a median 2.5 days of treatment. Most of them had MRC grade II disease (90%), and most presented with headache (96%), neck stiffness (96%), fever (85%), loss of consciousness (83%) and motor deficits (56%). Thirty three also had pulmonary involvement (69%), including 4 (12%) with miliary disease. All patients showed cerebrospinal fluid abnormalities typical of TB meningitis, and 34 (71%) had bacteriologically proven TBM based on microscopy (13/34), molecular testing (21/34) or culture (31/34). None of the cultured *M. tuberculosis* isolates were found to be rifampicin-resistant by GeneXpert.

At baseline, 45 out of 48 patients (94%) showed MRI abnormalities (Table 1), 40 (83%) had more than one brain lesion. The most common lesion was meningeal enhancement, mainly in the basal meninges and sylvian fissure (Fig 1A–1C). Tuberculomas were the second most common finding, most of them a miliary type (Fig 1D), while only one patient had a pseudo-abscess (Fig 1E1–1E4). On the contrary, brain infarctions were seen in 29 (60%) patients, mostly acute rather than chronic (Fig 1F1–1F3). Hydrocephalus was also common (56%), and always communicating, characterized by a broader callosal angle (Fig 1G), void signal appearance (Fig 1H), dilated temporal horn (Fig 1I), and larger of Evans' ratio (Fig 1J). Finally, 19% of our patients had cranial nerve imaging abnormalities (Fig 1K).

We next explored the relation between MRI findings and clinical characteristics. We divided into 5 groups of patient in order to weighted the brain abnormalities: normal, single, two, three, and four abnormalities. Patients with multiple brain abnormalities had a longer duration of illness, lower consciousness, higher numbers of motor deficits and cranial nerve palsy, more CSF abnormalities and higher positivity rate of CSF culture (Table 2). As only 4 (8.3%) of patients were HIV-infected, analysis was not stratified by HIV-status. Median time between onset of neurological symptoms and MRI (time to MRI) was 19 days with an interquartile of range 12–27. In univariate logistic regression analysis, time to MRI was not significantly associated with 6-month mortality (OR 1.02 [0.98–1.05], p = 0.315).

### MRI findings after two months drug treatment

After two months of treatment, one patient was not eligible for a second MRI because of insertion of a metal device for spinal TB, and one had movement artefacts and nine patients died before day 60. Among the remaining 37 patients, 89% (33/37) had new or worsening MRI findings. Only in 39% (13/33) of cases, this was accompanied by worsening clinical symptoms, the most common finding was new cranial nerve abnormalities. Two patients had worsening clinical symptoms without new MRI abnormalities: one patient developed a new central facial and hypoglossal nerve paresis at day 30, and another one had ptosis at day 30.

Among all patients with radiological worsening, 82% (27/33) had new or enlarged tuberculomas and 76% (25/33) had thicker or new location of meningeal enhancement.

(Fig 2). In addition, 24% (8/33) had new cranial nerve enhancement, one had developed a new infarction and two enlargements of hydrocephalus. In contrast to worsening meningeal enhancement and tuberculoma's, hydrocephalus was improving in 32% (12/37) of patients (Table 3).

**Table 1. Neuroradiological abnormalities at baseline.**

| | Tuberculous meningitis (n = 48) | |
|---|---|---|
| **Meningeal enhancement** | 39/48 (81%) | |
| Basal meninges | 25 | |
| Sylvian fissure | 22 | |
| Convexity | 18 | |
| Ventricular | 3 | |
| **Hydrocephalus** | 27/48 (56%) | |
| Communicans | 27 | |
| Non-communicans | 0 | |
| **Tuberculoma** | 37/48 (77%) | |
| Miliary | 35 | |
| Non-miliary | 24 | |
| Pseudo abscess | 1 | |
| **Brain Infarction** | 29/48 (60%) | |
| | Acute, n = 26 (90%) | Chronic, n = 4 (14%) |
| Cerebrum, basal ganglia **& thalamus** | 22 | 2 |
| Cerebrum, outside basal ganglia **& thalamus** | 22 | 2 |
| Cerebellum | 2 | 0 |
| Brainstem | 5 | 0 |
| **Cranial nerve enhancement** | 9/48 (19%) | |

Meningeal enhancement was defined as linear or nodular enhancement of meninges with contrast media [33] at one or more locations: the basal meninges (e.g. basal cistern, ambient cistern, quadrigeminal cistern, prepontine cistern, cerebellopontine cistern, suprasellar cistern, premedullary cistern), sylvian fissure, cerebral or cerebellar convexity/sulci and ventricular system [24]. Hydrocephalus was present if one or more of: dilated temporal horns of lateral ventricles, ballooning of frontal horns of lateral ventricle, ballooning of third ventricle, narrowed callosal angle, presence of flow void in T2W images at the Sylvian aquaducts [34]. Evans' index [35] of every patients with and without hydrocephalus were measured and compared. Communicating and non-communicating hydrocephalus were defined based on the absence or presence of an obstructing lesion along the intraventricular CSF pathways[34]. Tuberculomas were defined as the presence of nodular or ring enhancement with contrast media [33, 36] and specified as milliary (<2mm) or non-milliary (>2mm) tuberculomas, or pseudo-abscesses [37]. Abscesses were defined as the presence of ring enhancement with restricted diffusion appearance on DWI-ADC [38]. Acute infarctions were defined as lesions with restricted diffusion on DWI-ADC, and increased T2-weighted and fluid-attenuated inversion recovery (FLAIR) signal intensity [39]. Cranial nerve neuropathy was defined as enhancement with or without thickening of the oculomotor nerve, trigeminal nerve, abducens nerve, facial nerve or vestibulocochlear nerve [40].

## Neuroimaging in relation to patient outcome

Next, we examined MRI findings in relation to clinical outcome. As shown in Table 2, baseline MRI abnormalities did not associate to survival. When looking into the patients with paradoxical reactions at 2 months, 1 of 2 patients with clinical paradoxical reactions died, 0 in the group of radiological paradoxical reactions, and 2/13 (15%) in the group with clinical-radiological paradoxical reactions. However, the patients who had no paradoxical reactions or only had radiological changes without clinical worsening had more number of patient with good status recovery after 6 months of treatment (Table 4). These numbers were too small to perform formal statistics.

## Discussion

We studied neuroradiological abnormalities in a prospective and well-characterised cohort of TBM patients in Indonesia. Using 3D MP-RAGE imaging we found abnormalities in virtually

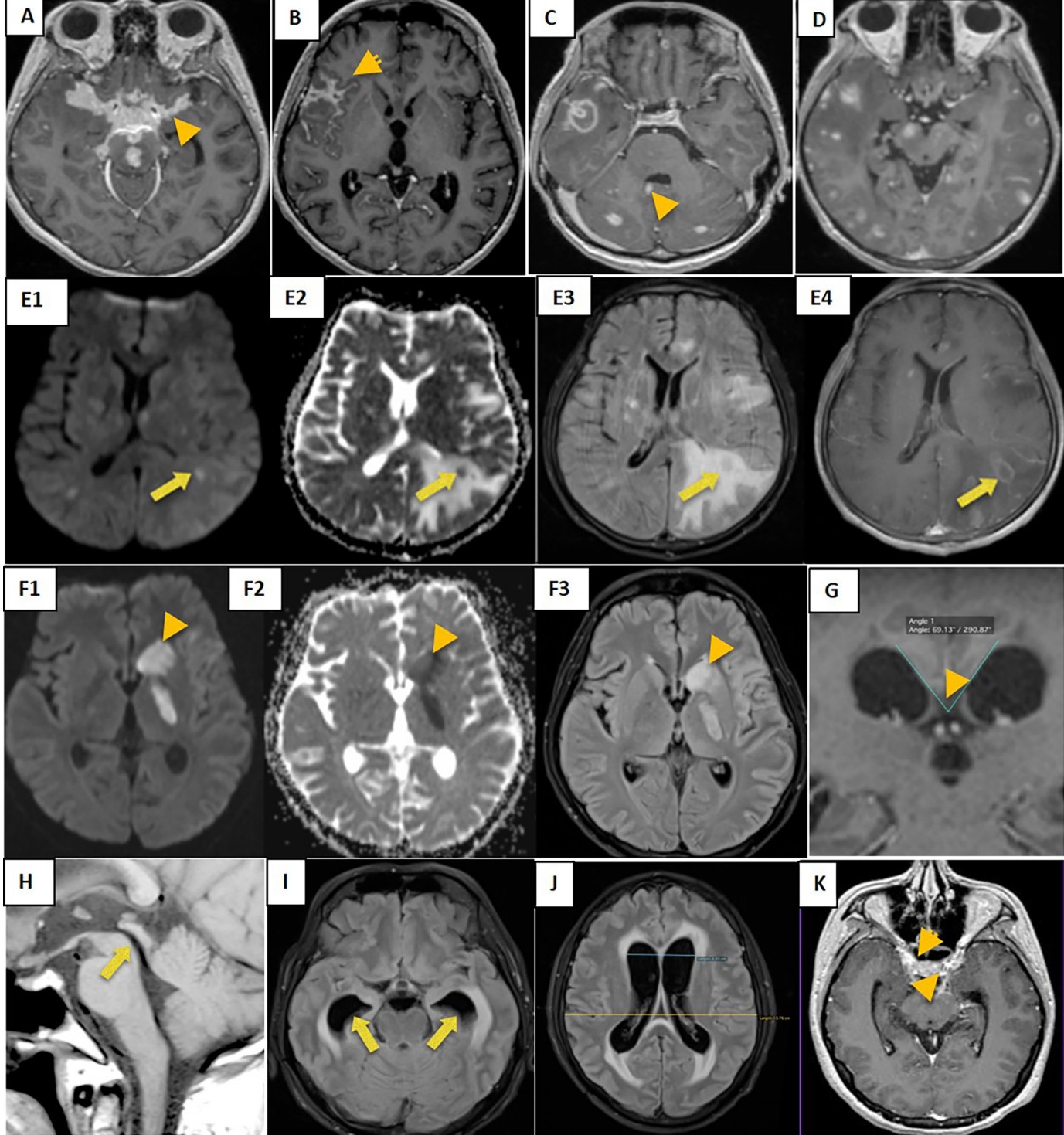

**Fig 1. Common baseline MRI findings in adults with tuberculous meningitis.** Meningeal enhancement at basal meninges (A), right Sylvian fissure (B), and ventricular (C); non-miliary and miliary tuberculomas (D); pseudo abscess in axial DWI (E1), axial ADC (E2), axial T2 FLAIR (E3) and axial T1W1 post contras (E4); multiple acute infarctions at the left basal ganglia in axial DWI (F1), axial ADC (F2), and axial T2 FLAIR (F3); Communicating hydrocephalus with narrowed Callosal angle (G), void signal in the aqueduct (H), and dilated temporal horn (I), broader of Evans' ratio (J); and oculomotor nerve enhancement (K).

**Table 2. Relation between disease characteristics and neuroimaging abnormalities.**

| | No Abnormality | single abnormality | two abnormalities | three abnormalities | four abnormalities |
|---|---|---|---|---|---|
| | **n = 3 (6%)** | **n = 5 (10%)** | **n = 9 (19%)** | **n = 15 (31%)** | **n = 16 (33%)** |
| Sex, Male | 2 (67) | 1 (20) | 5 (56) | 7 (47) | 8 (50) |
| Age, years | 19 [17–19] | 33 [25–47] | 36 [30–46] | 24 [20–33] | 32 [22–40] |
| Duration, days | 30 [7-] | 30 [10–90] | 8 [7–60] | 30 [14–90] | **33 [30–90]** |
| GCS | 15 | 14 [13–15] | 11 [11–13] | 13 [12–14] | **12 [12–13]** |
| Grade I | 0 (0) | 0 (0) | 0 (0) | 0 (0) | 0 (0) |
| II | 2 (67) | 5 (100) | 7 (78) | 14 (93) | 15 (94) |
| III | 1 (33) | 0 (0) | 2 (22) | 1 (7) | 1 (6) |
| Headache | 3 (100) | 5 (100) | 8 (89) | 14 (93) | 16 (100) |
| Neck stiffness | 3 (100) | 5 (100) | 9 (100) | 13 (87) | 16 (100) |
| Seizures | 1 (33) | 0 | 1 (11) | 1 (7) | 1 (6) |
| Motor deficits | 1 (33) | 1 (20) | 4 (44) | 9 (60) | **12 (75)** |
| Cranial nerve palsy | 1 (33) | 3 (60) | 7 (78) | **13 (87)** | **13 (81)** |
| Temperature ˚C | 38 [37-] | 37 [36–38] | 37 [37–38] | 38 [37–39] | 38 [37–38] |
| Fever | 1 (100) | 3 (60) | 6 (67) | 15 (100) | 14 (88) |
| Chest x-ray abnormalities | 0 | 1 (20) | 8 (89) | 10 (67) | **14 (88)** |
| Any bacteriological test positive | 0 | 3 (60) | 6 (67) | 11 (73) | **14 (88)** |
| CSF culture positivity | 0 | 3 (60) | 5 (56) | 10 (67) | **13 (81)** |
| HIV-infected | 1 (33) | 0 | 0 | 2 (13) | 1 (6) |
| 6-month mortality | 1 (33) | 1 (20) | 3 (33) | 3 (20) | 3 (19) |
| 6-month GOS | 0 | 1 (20) | 1 (11) | 0 | 3 (19) |
| Leukocytes (cells, µl) | 22 [8-] | 480 [224–763] | 199 [152–456] | 176 [84–215] | 258 [149–532] |
| PMN (cells/µl) | 326 [19-] | 52 [30–252] | 34 [1–160] | 57 [35–137] | 93 [149–532] |
| % of total CSF leukocytes | 14 | 25 [21–38] | 25 [21–71] | 31 [7–54] | 38 [18–61] |
| MN (cells/µl) | 137 [115-] | 155 [95–369] | 136 [7–332] | 110 [46–176] | 124 [67–371] |
| % of total CSF leukocytes | 86 [74-] | 75 [62–80] | 75 [29–80] | 69 [46–93] | 62 [39–371] |
| CSF protein (mg/dl) | 158 [32-] | 165 [89–182] | 209 [141–1434] | 188 [71–308] | **249 [160–402]** |
| CSF/blood glucose ratio | 0.22 [0.21-] | 0.25 [0.2–0.40 | 0.17 [0.13–0.24] | 0.25 [0.11–0.36] | **0.13 [0.09–0.19]** |
| Haemoglobin (mg/dL) | 14 [11-] | 13 [11–13] | 13 [10–15] | 11 [10–13] | 12 [10–14] |
| Leukocyte ($10^9$ cells/L) | 10 [7-] | 10 [6–13] | 1 [7–14] | 11 [9–14] | 11 [9–14] |
| Neutrophils (cells/ul) | 9 [6-] | 9 [4–10] | 10 [6–12] | 10 [7–12] | 9 [8–12] |
| % of total blood leukocytes | 84 [79-] | 81 [60–88] | 84 [79–86] | 84 [70–91] | 83 [79–88] |
| Lymphocytes ($10^9$ cells/L) | 0.9 [0.8-] | 0.9 [0.7–1.9] | 1.1 [0.8–1.6] | 1.2 [0.6–1.4] | 0.8 [0.8–1.1] |
| % of total blood leukocytes | 12 [9-] | 13 [8–21] | 9 [8–15] | 11 [5–21] | 8 [6–9] |
| Monocytes ($10^9$ cells/ul) | 0.5 (0.3-) | 0.6 (0.2–1.1) | 0.7 (0.5–1) | 0.7 (0.5–1) | 0.8 [0.4–1.0] |
| % of total blood leukocytes | 4 (3-) | 6 (4–13) | 6 (5–10) | 5 (4–10) | 8 [5–10] |
| Thrombocyte x $10^9$/L | 243 [210-] | 376 [287–425] | 290 [217–338] | 312 [239–468] | 479 [411–515] |
| Blood sodium (mEq/dL) | 135 [124-] | 127 [122–132] | 123 [116–136] | 124 [119–133] | 124 [116–129] |

Data are % of patients for categorical data or median value (IQR = interquartile range) for continuous data. **Abbreviations**: GCS, Glasgow Coma Scale; HIV, Human Immunodeficiency Virus; CSF, Cerebrospinal Fluid; PMN, Polymorphonuclear; MN, Mononuclear.

**Bold**: p-value <0.05 in comparison Krusskal-wallis test for numerical data and Chi-square test for categorical data. Fever: body temperature >38.5˚C.

all patients, with meningeal enhancement, tuberculomas, infarction, and hydrocephalus being the most frequent. Multiple MRI abnormalities were associated with more severe clinical presentations and more pronounced CSF abnormalities in an exploratory analysis. Many patients showed new or worsening MRI findings during the first two months of treatment, especially

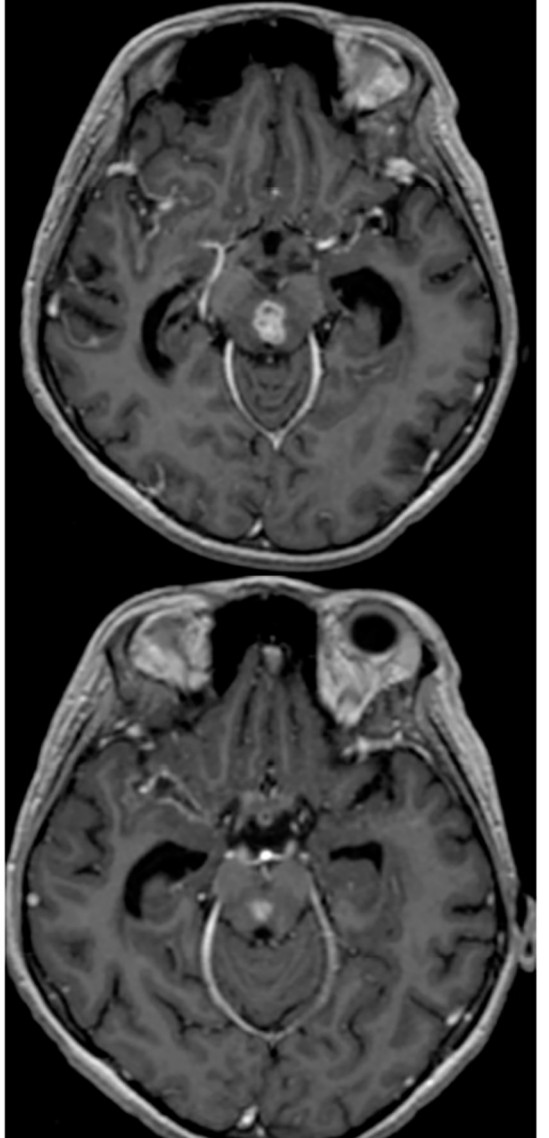
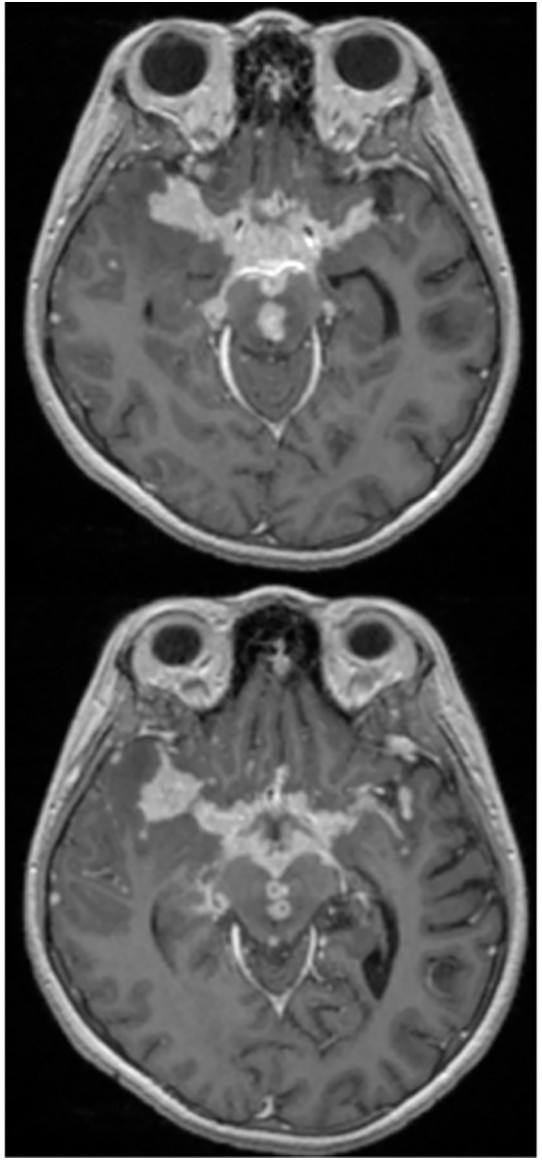

**Fig 2. Paradoxical response with basal meningeal enhancement after 60 days treatment.** Basal meningeal exudate before (**A**) and after two months after anti-tuberculosis drugs (**B**). Both from T1-W1 post contras from one patient with radiological worsening at day 60 days after anti-tuberculosis treatment.

**Table 3. Neuroimaging changes after two months of treatment in 37 patients.**

| Abnormality | Paradoxical worsening | No change | Improving |
|---|---|---|---|
| Meningeal enhancement | 25 | 9 | 3 |
| Tuberculoma | 27 | 8 | 2 |
| Hydrocephalus | 2 | 23 | 12 |
| Infarction | 1 | 36 | 0 |
| Cranial nerve neuropathy | 8 | 28 | 1 |

Paradoxical worsening was evaluated at day 60±1 week based on 37 paired MRI.

**Table 4. Outcome of paradoxical response.**

|  | Clinical and radiological (n = 13) | Clinical (n = 2) | Radiological (n = 20) | No Paradoxical response (n = 2) |
|---|---|---|---|---|
| 6-month mortality | 2 (15) | 1 (50) | 0 | 0 |
| 6-month good recovery | 7 (54) | 1 (50) | 20 (100) | 2 (100) |

Good recovery defined as score Glasgow Outcome Scale (GOS) 4 or 5.

miliary tuberculomas and meningeal enhancement. Although caution is warranted because of the size of the study, baseline MRI findings showed no significant association with the occurrence of paradoxical worsening, disability or death during follow-up.

Neuroradiological abnormalities in our setting were more common than in previous reports [18, 19]. This may be partially explained by the fact that TBM patients in this setting usually present late, with advanced disease [20]. The thick basal exudate that developed in the later stage of disease may entrap penetrating arteries and cranial nerves, and may block CSF resorption [2, 3]. Minimal lesions in early disease may be missed on conventional spin-echo images, especially if acquired without contrast [21]. Another possible explanation for the difference with previous reports is that we had a high proportion of microbiologically confirmed TBM, which we have previously found to be associated with inflammation [22]. The three patients without abnormalities on brain MRI in this study also had negative microbiological test results.

Our finding match with earlier observations, which showed that most TBM patients present with multiple baseline MRI abnormalities [23, 24], predominantly meningeal enhancement and tuberculoma [6, 21, 24–26]. Meningeal enhancement is mainly found at the basal subarachnoid cisterns. Dense gelatinous leptomeningeal exudates were the commonest pathology in TBM [20, 27]. In our patients, more extensive brain abnormalities were associated with a lower GCS, motor and cranial nerve palsies, presence of lung TB, microbiological confirmation, and a lower CSF glucose and higher CSF protein, reflecting inflammation or more severe disease. Our result differs from a study in Vietnam that did not find any association between baseline MRI findings on admission with most of clinical or CSF inflammatory markers [24].

The majority of our patients had tuberculomas at baseline, much more than that in previous studies [24–26]. Tuberculomas were predominantly of a miliary type, match those observed in earlier studies [23, 24]. The higher proportion of tuberculomas in our study may be due to the use of the sensitive 3D MP-RAGE. The 3D MP-RAGE sequence used in our protocol produces slices with a thickness of 1 mm without any gap. This allows for robust detection of brain lesions, including small tuberculomas and minimal meningeal enhancement.

Newly developing or enlarging intracranial tuberculomas following initial improvement may be observed despite appropriate anti-tuberculosis therapy [28, 29]. Many of our patients indeed showed worsening of MRI findings (so-called 'paradoxical reactions') after two months anti-tuberculosis treatment, with new miliary tuberculomas and meningeal enhancement being the most common. Interestingly, the majority of these radiological paradoxical reactions were not accompanied by new clinical findings. The present finding is consistent with other studies that indicate that the most common radiological worsening was an asymptomatic increase in number or size of tuberculomas [24, 30]; mostly occurring within 3 months of initiating anti-tuberculosis treatment [30].

In agreement with previous studies in India [25, 31] and Vietnam [24], MRI appearances at baseline and two months were not associated with death or severe disability after six months of the treatment, although this part of our analysis was only exploratory and may be underpowered. Indeed, in larger studies, hydrocephalus [32] and infarction [20] significantly predicted

TBM mortality. In accordance with our result, previous studies have demonstrated that paradoxical reactions did not predict 6-month mortality [30]. In our study, brain infarctions and tuberculomas were mostly small without a mass-effect leading to brain oedema. In line with earlier studies [3, 4, 21] we only found one case of tubercular abscess; a large solitary lesion with ring enhancement and surrounding vasogenic oedema and mass effect. However, in contrast to earlier findings [31], we found a better functional recovery among patients without paradoxical reactions, presumably caused by preservation of corticospinal and corticobulbar tract that might be affected by the ischaemic damage of new or enlarged tuberculoma. In this study, patients with tuberculoma's more often presented with motor and cranial nerve abnormalities compared to those without tuberculomas regardless of other brain lesion (p = 0.032 and 0.001). It is important that the finding of a paradoxical reactions is not interpreted as treatment failure, presence of an alternative diagnosis or drug-resistance as this may lead to hazardous decision of TB-drug cessation.

Strengths of our study are an extensive descriptive analysis facilitated by 3D MP-RAGE in the MRI protocol, in a well-described clinical cohort, with high proportion of microbiologically confirmed disease (71%). However, the study's sample size limits the possibility of drawing definite conclusions in correlating neuroradiological findings to clinical parameters. Also, our study was not designed to help identify TBM-specific abnormalities as it lacked a control group with patients suffering different brain infections.

In conclusion, MRI abnormalities in this group of TBM patients were common, and paradoxical reactions often occurred despite use of steroids. This result provides further support for using neuroradiological imaging in TBM diagnosis and treatment evaluation, including the evaluation of paradoxical reactions. Also, MRIs could help characterize immunopathology in drug trials or studies focusing on development of new host-directed strategies for TBM.

## Supporting information

**S1 Raw data.**
(XLSX)

## Author Contributions

**Conceptualization:** Sofiati Dian, Robby Hermawan, Ahmad Rizal Ganiem, Reinout van Crevel.

**Data curation:** Sofiati Dian, Robby Hermawan, Farhan Anwary, Ristaniah D. Soetikno.

**Formal analysis:** Sofiati Dian.

**Funding acquisition:** Sofiati Dian, Tri Hanggono Achmad, Rovina Ruslami.

**Investigation:** Sofiati Dian, Sofia Immaculata, Farhan Anwary.

**Methodology:** Sofiati Dian, Arjan van Laarhoven, Reinout van Crevel.

**Project administration:** Sofia Immaculata.

**Resources:** Ristaniah D. Soetikno.

**Software:** Robby Hermawan.

**Supervision:** Tri Hanggono Achmad, Rovina Ruslami, Ahmad Rizal Ganiem.

**Validation:** Reinout van Crevel.

**Writing – original draft:** Sofiati Dian.

**Writing – review & editing:** Robby Hermawan, Arjan van Laarhoven, Ahmad Rizal Ganiem, Reinout van Crevel.

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
