## [Decision Letter · Decision Letter 0]

24 Mar 2020

PONE-D-19-33449

Brain MRI findings in relation to clinical characteristics and outcome of tuberculous meningitis

PLOS ONE

Dear dr Dian,

Thank you for submitting your manuscript to PLOS ONE. After careful consideration, we feel that it has merit but does not fully meet PLOS ONE’s publication criteria as it currently stands. Therefore, we invite you to submit a revised version of the manuscript that addresses the points raised during the review process.

I would also like to apologize for the time that it has taken to render a decision on your manuscript. I have had a very difficult time securing a second reviewer. However, I believe that the comments that have been received will be valuable in improving the quality of the manuscript. Please carefully address all of the comments that have been raised, or explicitly state why this has not been done. In addition, I have the following points that should be addressed:

1. Line 135: Please change "Swiss" to "Switzerland"

2. Line 143: You state that the basal ganglia was considered as "caudate nucleus, internal capsule, lentiform nucleus and thalamus." This is incorrect. The thalamus is not part of the basal ganglia, and there other brain structures that are part of the basal ganglia, but not listed here (e.g. nucleus accumbens, olfactory tubercle, substantia nigra, and subthalamic nucleus.). There appears to be some confusion here based on the cited reference (Tai, et al. Scientific Reports volume 6, Article number: 38802 (2016)) . Please clarify.

3. Line 251: It is a bit misleading to refer to a "thin gap" in the 3D MP-RAGE sequence. As it is a 3D sequence, there is no gap at all.

We would appreciate receiving your revised manuscript by May 08 2020 11:59PM. To enhance the reproducibility of your results, we recommend that if applicable you deposit your laboratory protocols in protocols.io, where a protocol can be assigned its own identifier (DOI) such that it can be cited independently in the future. For instructions see: http://journals.plos.org/plosone/s/submission-guidelines#loc-laboratory-protocols

We look forward to receiving your revised manuscript.

Kind regards,

Niels Bergsland

Academic Editor

PLOS ONE

Journal Requirements:

2. Please include the registration number for the clinical trial referenced in the manuscript.

Reviewers' comments:

Reviewer's Responses to Questions

**Comments to the Author**

1. Is the manuscript technically sound, and do the data support the conclusions?

Reviewer #1: Yes

2. Has the statistical analysis been performed appropriately and rigorously? 

Reviewer #1: I Don't Know

3. Have the authors made all data underlying the findings in their manuscript fully available?

Reviewer #1: Yes

4. Is the manuscript presented in an intelligible fashion and written in standard English?

Reviewer #1: Yes

5. Review Comments to the Author

Reviewer #1: I read carefully the manuscript PONE-D-19-33449

First i will like thanks authors for the quality of the manuscript which is easy to read and well organized

This topic in Tuberculosis disease is rare and recent datas are needed to improve the management of such patient world wide.

The authors through their MRI focus reinforce previously published message on MRI major place as neurological diagnostic tool for TB.

Major concern

- Could the authors give information on time between first symptoms and and first MRI ? It might be more relevant than delay between treatment begining and MRI.

Line 196 "Only in 39% (13/30) of cases..." I do not understand how you find 30 because just before you speak about 33 cases, please fix it.

- Lines 196-1977 the sentence " this was accompanied by worsening clinical symptoms, the most common finding being …" is incomplete, so please complete it ?

- What was the dexamethasone posology and what tappering paln do you use ?

- What have you done with dexamethasone dosis when worsening was proved clinically and/or radiologically ?

- What were the pulmonary findings for these patients (Chest X-Ray and sputum samples) and have you seen association between type of lung involvement and neurological loacalization of TB ?

- You stopped follow up at M6 but neurological tuberculosis need up to 9 months of treatment. Could you argue why a such outcome? Have you datas on M6-M12 periods?

Minor

There is some publication in western country on this topic you could cite :

- A. Bleibtreu et al. / Médecine et maladies infectieuses 48 (2018) 533–539

-Venkatraman N, King T, Bell D, Woltmann G, Wiselka M, AbubakarI, et al. High levels of neurological involvement but low mortalityin miliary tuberculosis: a 6-year case-series from the UK. Eur RespirJ 2016;47(5):1578–81

6. PLOS authors have the option to publish the peer review history of their article (what does this mean?). If published, this will include your full peer review and any attached files.

Reviewer #1: Yes: Dr Bleibtreu Alexandre MD, PhD. Infectious diseases Specialist

---

## [Author Response · Author response to Decision Letter 0]

4 May 2020

Authors’ Response to the Review Comments

Journal Plos One

Manuscript# PONE-D-19-33449

Title of Paper Brain MRI findings in relation to clinical characteristics and outcome of tuberculous meningitis

Authors Sofiati Dian, Robby Hermawan, Arjan van Laarhoven, Sofia Immaculata, Tri Hanggono Achmad, Rovina Ruslami, Farhan, , Ristaniah D. Soetikno, Ahmad Rizal Ganiem, Reinout van Crevel 

We appreciate the time and efforts by the editor and referees in reviewing this manuscript. We have addressed all issues in the review report and hope that our revised manuscript is now suitable for publication in Plos One.

Response to Comments from Academic Editor

Comments:

I would also like to apologize for the time that it has taken to render a decision on your manuscript. I have had a very difficult time securing a second reviewer. However, I believe that the comments that have been received will be valuable in improving the quality of the manuscript.

We would like to thank Dr. Bergsland as the academic editor for the positive appraisal of our manuscript. We greatly appreciate your efforts to carefully review the paper and the valuable suggestions offered.

Suggestions:

1) Line 135: Please change "Swiss" to "Switzerland” (Line 136)

Done

2) Line 143: You state that the basal ganglia were considered as "caudate nucleus, internal capsule, lentiform nucleus and thalamus." This is incorrect. The thalamus is not part of the basal ganglia, and there other brain structures that are part of the basal ganglia, but not listed here (e.g. nucleus accumbens, olfactory tubercle, substantia nigra, and subthalamic nucleus.). There appears to be some confusion here based on the cited reference (Tai, et al. Scientific Reports volume 6, Article number: 38802 (2016)). Please clarify.

We thank the editor for the sharp attention on this matter. Indeed, we made an unclear statement about the categorization and have already made a correction on it.

Original sentence:

Cerebral infarction was further categorized into inside or outside the basal ganglia. Infarction was categorized into basal ganglia if located in caudate nucleus, internal capsule, lentiform nucleus and thalamus. 

New sentence:

Cerebral infarction was further categorized into inside or outside the basal ganglia and thalamus. Infarction was categorized as in the basal ganglia if located in the striatum or caudate nucleus, putamen, and globus palidus.(1) (Line 142-144, table 1 below category “Brain Infarction”)

We put thalamus and basal ganglia into one category as due to their vascularization are shared similar anatomical feature and are similarly affected by vasculitis. TB meningitis may induce vasculitis of small and medium-sized cerebral arteries, often the lenticulostriate arteries or the posterior cerebral branches and the thalamoperforating arteries that send their blood supply to basal ganglia and thalamus respectively.(2)

In the definition of basal ganglia, we did not mention the nucleus accumbent and olfactory tubercle because both structures are too small to be detected reliably by MRI.(1) The substantia nigra and subthalamic nucleus were excluded as they are located in the midbrain and are not part of cerebral structure.

2) Line 251: It is a bit misleading to refer to a "thin gap" in the 3D MP-RAGE sequence. As it is a 3D sequence, there is no gap at all.

The editor’s statement is very relevant, indeed we intended to mention “thickness” and not “gap” in this sentence. We have rephrased this sentence. 

Original sentence: 

The thin gap between slices in our protocol allowed a robust detection of brain lesion, including tuberculomas and meningeal enhancement.

New sentence:

The 3D MP-RAGE sequence used in our protocol produces slices with a thickness of 1 mm without any gap. This allows for robust detection of brain lesions, including small tuberculomas and minimal meningeal enhancement. (Line 263-265)

 

Response to Comments from Reviewer 

Comments:

First, I will like thanks authors for the quality of the manuscript which is easy to read and well organized. This topic in Tuberculosis disease is rare and recent data are needed to improve the management of such patient worldwide. The authors through their MRI focus reinforce previously published message on MRI major place as neurological diagnostic tool for TB. 

We thank also this reviewer for his/her compliments and constructive review.

Major concern:

Comments: 

1. Could the authors give information on time between first symptoms and first MRI? It might be more relevant than delay between treatment beginning and MRI.

It should be noted that MRI was not used for diagnosis of TB meningitis in our setting. Still, we have added the time between onset of neurological symptoms and MRI (median was 19 days with an interquartile of range 12-27. In univariate logistic regression analysis, time to MRI was not significantly associated with 6-month mortality (OR 1.02 [0.98-1.05], p=0.315). (Line 194-197)

2. Line 196 "Only in 39% (13/30) of cases..." I do not understand how you find 30 because just before you speak about 33 cases, please fix it.

Thank you for pointing out this mistake, it was a typo and we have corrected the denominator in the newer version of the manuscript. (Line 204)

3. Lines 196-1977 the sentence " this was accompanied by worsening clinical symptoms, the most common finding being …" is incomplete, so please complete it?

The reviewer is correct, we have completed the sentence in the newer version of the manuscript:

Only in 39% (13/33) of cases, this was accompanied by worsening clinical symptoms, the most common finding was new cranial nerve abnormalities. Two patients had worsening clinical symptoms without new MRI abnormalities: one patient developed a new central facial and hypoglossal nerve palsy at day 30, and another one had ptosis at day 30. (Line 204-207)

4. What was the dexamethasone posology and what tapering plan do you use? 

We adopted the doses of dexamethasone for tuberculous meningitis accordingly to the randomized controlled trial by Thwaites, 2004.(3) (We added the reference in Line 106)

Patients with grade I of BMRC: 

- Week 1: 0.3 mg per kilogram per day intravenously

- Week 2: 0.2 mg per kilogram per day intravenously 

- Week 3: oral treatment at a total of 0.1 mg per kilogram per day

- Week 4: oral treatment at a total of 3 mg per day

- Week 5: oral treatment at a total of 2 mg per day

- Week 6: oral treatment at a total of 1 mg per day

Patients with grade II or III: 

- Week 1: 0.4 mg per kilogram per day intravenously

- Week 2: 0.3 mg per kilogram per day intravenously

- Week 3: 0.2 mg per kilogram per day intravenously

- Week 4: 0.1 mg per kilogram per day intravenously

- Week 5: Oral treatment at a total of 4 mg per day 

- Week 6: Oral treatment at a total of 3 mg per day 

- Week 7: Oral treatment at a total of 2 mg per day 

- Week 8: Oral treatment at a total of 1 mg per day 

In the case of early discharge, intra venous dexamethasone will be substituted with oral preparation, given in similar dose to continue dexamethasone administration until its completion, or switch to equivalent dose of methylprednisolone.

Equivalent dose: Dexamethasone 2mg ~ Methylprednisolone 12mg

5. What have you done with dexamethasone doses when worsening was proved clinically and/or radiologically?

Thank you for this question, we have rephrased the sentence about this to make it clearer. 

Original sentence:

Corticosteroids were given to patients with symptomatic paradoxical response. (Line 114)

New sentence:

In case the patient experienced a paradoxical response, we increased the dose of dexamethasone back to the starting dose accordingly to their TBM grade. (Line 110-112)

6. What were the pulmonary findings for these patients (Chest X-Ray and sputum samples) and have you seen association between type of lung involvement and neurological localization of TB? 

In this study, we found 33 (69%) patients with pulmonary TB, including 4 (12%) with miliary disease (lines 176-177). Among those 4, 2 had brain infarction in the basal ganglia and thalamus, 3 had brain infarction outside basal ganglia and thalamus, and 2 had brain infarction in the cerebellum and brain stem. Among 33 patients with pulmonary TB, 19 (58%) had brain infarction in the basal ganglia and thalamus, and only 2 (6%) and 3 (9%) had cerebellum or brainstem infarction. The patients who had normal chest x-ray (n=15) were mostly had no brain infarction 10/15 (67%) but had quite high proportion of tuberculoma 10/15 (67%). Of note, none of the patients had single location lesion (infarction or tuberculoma). We failed to find the association between type of this lung involvement with the neurological localization on TB.

6. You stopped follow up at M6 but neurological tuberculosis needs up to 9 months of treatment. Could you argue why a such outcome? Have you data on M6-M12 periods?

Minor

At the time of this randomized clinical trial, the national TB guideline in Indonesia recommended 6 months treatment.(4) We did not specifically collect data beyond 6 months of follow-up. 

7. There is some publication in western country on this topic you could cite:

- A. Bleibtreu et al. / Médecine et maladies infectieuses 48 (2018) 533–539

-Venkatraman N, King T, Bell D, Woltmann G, Wiselka M, AbubakarI, et al. High levels of neurological involvement but low mortality in miliary tuberculosis: a 6-year case-series from the UK. Eur Respir J 2016;47(5):1578–81

Thank you for the references. We have read the publication with a great interest. Both articles recommend the neuroimaging assessment in patients with lung miliary tuberculosis (TB), since they found high incidence of brain lesions among patients with miliary TB, despite no neurological symptoms. We personally agree with this recommendation. However, since our study included patients with clinical and CSF findings compatible to tuberculous meningitis, we could not relate this finding with our findings.

Reference

1. Lanciego JL, Luquin N, Obeso JA. Functional neuroanatomy of the basal ganglia. Cold Spring Harb Perspect Med. 2012;2(12):a009621.

2. Abdel Razek AA, Alvarez H, Bagg S, Refaat S, Castillo M. Imaging spectrum of CNS vasculitis. Radiographics. 2014;34(4):873-94.

3. Thwaites GE, Nguyen DB, Nguyen HD, Hoang TQ, Do TT, Nguyen TC, et al. Dexamethasone for the treatment of tuberculous meningitis in adolescents and adults. N Engl J Med.2004;351(17):1741-51.

4. WHO. Treatment of Tuberculosis guidelines. Geneva, Switzerland.2010.

1. We have revised this manuscript as requested accordingly to the PLOS ONE's style requirements, including those for file naming.

2. We have included the registration number for the clinical trial referenced in the manuscript.

3. Upon the repository information, all relevant data are within the manuscript and additional data will provide when asked.

---

## [Decision Letter · Decision Letter 1]

19 Jun 2020

PONE-D-19-33449R1

Brain MRI findings in relation to clinical characteristics and outcome of tuberculous meningitis

PLOS ONE

Dear Dr. Dian,

Thank you for submitting your manuscript to PLOS ONE. After careful consideration, we have decided that your manuscript does not meet our criteria for publication and must therefore be rejected.

I recognize that you have waited for a long time to have a decision on the manuscript. The manuscript was sent to an additional reviewer who specifically assessed the statistical analysis. The reviewer had major concerns about the statistical approach, resulting in severe doubts about the validity of the results.

I am sorry that we cannot be more positive on this occasion, but hope that you appreciate the reasons for this decision.

Yours sincerely,

Niels Bergsland

Academic Editor

PLOS ONE

Reviewers' comments:

Reviewer's Responses to Questions

**Comments to the Author**

1. If the authors have adequately addressed your comments raised in a previous round of review and you feel that this manuscript is now acceptable for publication, you may indicate that here to bypass the “Comments to the Author” section, enter your conflict of interest statement in the “Confidential to Editor” section, and submit your "Accept" recommendation.

Reviewer #1: All comments have been addressed

Reviewer #2: (No Response)

2. Is the manuscript technically sound, and do the data support the conclusions?

Reviewer #1: Yes

Reviewer #2: No

3. Has the statistical analysis been performed appropriately and rigorously? 

Reviewer #1: Yes

Reviewer #2: No

4. Have the authors made all data underlying the findings in their manuscript fully available?

Reviewer #1: Yes

Reviewer #2: Yes

5. Is the manuscript presented in an intelligible fashion and written in standard English?

Reviewer #1: Yes

Reviewer #2: No

6. Review Comments to the Author

Reviewer #1: All concerns were adressed by the authors. The response to reviewers concerns is point by point argue and clear.

I am agree with their decision to not improved the references.

For me the manuscript is suitable for publication

Reviewer #2: In this study, the authors used MRI data collected at baseline in 48 patients with tuberculous meningitis (TBM). They then attempted to test the association between MRI findings (which included meningeal enhancement, tuberculomas, brain infarctions, and hydrocephalus).and clinical characteristics (motor deficits, cranial nerve palsy, microbiological confirmation, and CSF abnormalities). This analysis, however is flawed because none of these comparisons are “clean.” This is due to the fact that so many of the patients had multiple MRI abnormalities. It doesn’t make sense to test each abnormality separately against the others and make conclusions about an individual abnormality given that there are a large number of possible permutations of abnormalities that could occur in any given patient. There is no way to be certain that any one abnormality is associated with any clinical characteristic, since the abnormalities so frequently occur in combination.

The way to do this correctly would be to create a model in which the presence or absence of the clinical characteristic is the outcome, and a binary variable is created for each of the potential MRI abnormalities (allowing for one subject to have multiple abnormalities). It is doubtful that there are enough data to give sufficient power to such an analysis. Furthermore, it would also be advisable to consider interactions between the MRI findings (e.g., do patients with both meningeal enhancement AND tuberculomas have a different relationship with a clinical finding than patients who have meningeal enhancement WITHOUT tuberculomas). The small sample would clearly lack sufficient power to see any sort of interactions. I don’t think there is any way this can be fixed.

The same problem is inherent in the analysis of the effect of brain abnormalities on six month mortality and functional outcome. It isn’t correct to examine each one separately, for the above reasons. A model would need to contain a variable (yes/no) for EACH abnormality. One possible option would be to count the number of brain abnormalities any patient had, and check whether the number of abnormalities was associated with the different clinical outcomes. I’m not sure this would yield useful information, however.

It is not surprising that the authors found none of the MRI abnormalities (again, examined separately) were associated with mortality, disability, or worsening. There is too much overlap between any “one” abnormality and “all others,” since very few subjects HAD just one abnormality. Again, it requires a much more complex analysis than is possible given the small size of the data.

I feel that the only way these data could be presented would be descriptively, without an attempt to derive conclusions from very oversimplified (and incorrect) analyses.

Minor points:

The authors state that 60 patients were enrolled in the “parent” study (a Phase 2b dose-finding study). That is true, but if 12 of these did not have a baseline MRI, then they would never have been considered for the current study. A better way to present this would be to say that of 60 patients enrolled in a Phase 2b dose-finding clinical trial, 48 had baseline MRIs and were therefore eligible for the current study.

l.162 Multivariable, not multivariate

Survival analysis, not logistic regression, should be used to look at six-month mortality, but again, the fact that subjects could have multiple MRI abnormalities would have to be accounted for.

7. PLOS authors have the option to publish the peer review history of their article (what does this mean?). If published, this will include your full peer review and any attached files.

Reviewer #1: Yes: Bleibtreu Alexandre

Reviewer #2: No

- - - - -

---

## [Author Response · Author response to Decision Letter 1]

28 Sep 2020

Authors’ Response to the Review Comments

Journal Plos One

Manuscript# PONE-D-19-33449R1

Title of Paper Brain MRI findings in relation to clinical characteristics and outcome of tuberculous meningitis

Authors Sofiati Dian, Robby Hermawan, Arjan van Laarhoven, Sofia Immaculata, Tri Hanggono Achmad, Rovina Ruslami, Farhan Anwary, Ristaniah D. Soetikno, Ahmad Rizal Ganiem, Reinout van Crevel

Response to Comments from Reviewer #1

Comments:

All concerns were addressed by the authors. The response to reviewers’ concerns is point by point argue and clear. I am agree with their decision to not improved the references.

For me the manuscript is suitable for publication.

We would like to thank Dr. Bleibtreu for the positive appraisal of our manuscript. 

Response to Comments from Reviewer #2

Comment 1: 

In this study, the authors used MRI data collected at baseline in 48 patients with tuberculous meningitis (TBM). They then attempted to test the association between MRI findings (which included meningeal enhancement, tuberculomas, brain infarctions, and hydrocephalus).and clinical characteristics (motor deficits, cranial nerve palsy, microbiological confirmation, and CSF abnormalities). This analysis, however is flawed because none of these comparisons are “clean.” This is due to the fact that so many of the patients had multiple MRI abnormalities. It doesn’t make sense to test each abnormality separately against the others and make conclusions about an individual abnormality given that there are a large number of possible permutations of abnormalities that could occur in any given patient. There is no way to be certain that any one abnormality is associated with any clinical characteristic, since the abnormalities so frequently occur in combination.

Response 1:

Thanks for including a separate statistical review, bringing up relevant point. Indeed many of the associations in these clinical studies are ‘unclean’, as pointed out, i.e. in previous report by Thwaites et.al, Kalita et.al, and Wasay et.al, many events occur together. Moreover, due to its relative infrequent nature, series of tuberculous meningitis patients are often small. Among all typical brain MRI findings, none has consistently been linked with a bad outcome. Therefore, to make it sure, we weighted the abnormalities based on the total number of it: ‘no abnormalities’, a single abnormality, or 2, 3 or 4 abnormalities (table 2), showing that the presence of 4 abnormalities is associated with more severe clinical presentation, bacteriological confirmation and CSF inflammation.

Comment 2:

The way to do this correctly would be to create a model in which the presence or absence of the clinical characteristic is the outcome, and a binary variable is created for each of the potential MRI abnormalities (allowing for one subject to have multiple abnormalities). It is doubtful that there are enough data to give sufficient power to such an analysis. Furthermore, it would also be advisable to consider interactions between the MRI findings (e.g., do patients with both meningeal enhancement AND tuberculomas have a different relationship with a clinical finding than patients who have meningeal enhancement WITHOUT tuberculomas). The small sample would clearly lack sufficient power to see any sort of interactions. I don’t think there is any way this can be fixed.

Response 2:

We agree with the reviewer that sample size is a concern (line 286). Still, this study represents serial MRI analysis of the 2nd largest cohort of well characterized and very high proportion of bacteriologically proven disease of TBM patients, a disease with a 1% incidence among human tuberculosis patients. In response to the reviewer’s comment, we now stress more clearly that the associative analysis of neuroradiology findings with the clinical and laboratory features is of an explorative nature (line 50, 89, 186, 227, 269). We also incorporated the suggestion of weighting the brain abnormality by the number of the finding (table 2).

Comment 3:

The same problem is inherent in the analysis of the effect of brain abnormalities on six-month mortality and functional outcome. It isn’t correct to examine each one separately, for the above reasons. A model would need to contain a variable (yes/no) for EACH abnormality. One possible option would be to count the number of brain abnormalities any patient had, and check whether the number of abnormalities was associated with the different clinical outcomes. I’m not sure this would yield useful information, however.

Response 3:

Regarding this study more as descriptive and exploratory, a complete statistical model – also given the study size – might not be appropriate. However, following the reviewer’s suggestion we have examined a possible association between the number of MRI abnormalities, paradoxical response and patient outcome (table 2 and table 4). This shows a comparable number of patients who died or had a good recovery after 6-month of treatment among groups with different number of brain abnormalities. However, good recovery after 6-month of treatment seems more common in group of patients with only radiological or had no paradoxical response.

Comment 4:

It is not surprising that the authors found none of the MRI abnormalities (again, examined separately) were associated with mortality, disability, or worsening. There is too much overlap between any “one” abnormality and “all others,” since very few subjects HAD just one abnormality. Again, it requires a much more complex analysis than is possible given the small size of the data.

I feel that the only way these data could be presented would be descriptively, without an attempt to derive conclusions from very oversimplified (and incorrect) analyses.

Response 4:

Following the reviewer’s suggestion, we now present the data in a more descriptive way. We have rephrased several statements to make it less strong in the abstract (lines 50), introduction (line 86), results (lines 186) and discussion (lines 227, 269):

Minor points:

The authors state that 60 patients were enrolled in the “parent” study (a Phase 2b dose-finding study). That is true, but if 12 of these did not have a baseline MRI, then they would never have been considered for the current study. A better way to present this would be to say that of 60 patients enrolled in a Phase 2b dose-finding clinical trial, 48 had baseline MRIs and were therefore eligible for the current study.

Response minor point:

Thanks for this suggestion, we have rephrased this sentence (line 167-171).

Comment 5

l.162 Multivariable, not multivariate

Response 5:

Thank you for pointing out this mistake; we deleted this sentence altogether because we changed the analysis.

Comment 6:

Survival analysis, not logistic regression, should be used to look at six-month mortality, but again, the fact that subjects could have multiple MRI abnormalities would have to be accounted for.

Response 6:

We thank the reviewer for pointing this out. Indeed, it will be less accurate to predict the mortality by cox or logistic model using this single brain lesion due to the ‘unclean’ variable of each brain abnormality as pointed out by the reviewer. As suggested and a previously applied (Kalita, et.al) we adopted to a descriptive approach (table 2). in a larger study, this approach might have been used (for instance as done by Wasay et.al(3) in their study), but we have now used a more descriptive approach as Kalita, et.al(2) have done (table 2).

We again thank the editor and reviewers for their time and efforts to help improve our manuscript which we hope is now suitable for publication.

 

Reference

1. Thwaites GE, Macmullen-Price J, Tran TH, Pham PM, Nguyen TD, Simmons CP, et al. Serial MRI to determine the effect of dexamethasone on the cerebral pathology of tuberculous meningitis: an observational study. Lancet Neurol. 2007;6(3):230-6.

2. Kalita J, Prasad S, Misra UK. Predictors of paradoxical tuberculoma in tuberculous meningitis. Int J Tuberc Lung Dis. 2014;18(4):486-91.

3. Wasay M, Farooq S, Khowaja ZA, Bawa ZA, Ali SM, Awan S, et al. Cerebral infarction and tuberculoma in central nervous system tuberculosis: frequency and prognostic implications. J Neurol Neurosurg Psychiatry. 2014;85(11):1260-4.

4. Anuradha HK, Garg RK, Agarwal A, Sinha MK, Verma R, Singh MK, et al. Predictors of stroke in patients of tuberculous meningitis and its effect on the outcome. QJM. 2010;103(9):671-8.

5. Singh AK, Malhotra HS, Garg RK, Jain A, Kumar N, Kohli N, et al. Paradoxical reaction in tuberculous meningitis: presentation, predictors and impact on prognosis. BMC Infect Dis. 2016;16(1):306.

---

## [Editor Report · Decision Letter 2]

26 Oct 2020

Brain MRI findings in relation to clinical characteristics and outcome of tuberculous meningitis

PONE-D-19-33449R2

Dear Dr. Dian,

First, I would like to thank you for your enormous patience during this entire process!

We’re pleased to inform you that your manuscript has been judged scientifically suitable for publication and will be formally accepted for publication once it meets all outstanding technical requirements.

Kind regards,

Niels Bergsland

Academic Editor

PLOS ONE
---

## [Editor Report · Acceptance letter]

4 Nov 2020

PONE-D-19-33449R2 

Brain MRI findings in relation to clinical characteristics and outcome of tuberculous meningitis 

Dear Dr. Dian:

I'm pleased to inform you that your manuscript has been deemed suitable for publication in PLOS ONE. Congratulations! Your manuscript is now with our production department. 

Kind regards, 

on behalf of

Dr. Niels Bergsland 

Academic Editor

PLOS ONE